# Hyper-cores promote localization and efficient seeding in higher-order processes

Marco Mancastroppa [1], Iacopo Iacopini [2,3], Giovanni Petri [2,4] & Alain Barrat [1]✉

Going beyond networks, to include higher-order interactions of arbitrary sizes, is a major step to better describe complex systems. In the resulting hypergraph representation, tools to identify structures and central nodes are scarce. We consider the decomposition of a hypergraph in hyper-cores, subsets of nodes connected by at least a certain number of hyperedges of at least a certain size. We show that this provides a fingerprint for data described by hypergraphs and suggests a novel notion of centrality, the hypercoreness. We assess the role of hyper-cores and nodes with large hypercoreness in higher-order dynamical processes: such nodes have large spreading power and spreading processes are localized in central hyper-cores. Additionally, in the emergence of social conventions very few committed individuals with high hypercoreness can rapidly overturn a majority convention. Our work opens multiple research avenues, from comparing empirical data to model validation and study of temporally varying hypergraphs.

Network theory provides a powerful framework to describe a wide range of complex systems whose elements interact in pairs[1-4]: this theory has developed numerous concepts and techniques to characterize the structure of complex networks at various scales, from the single element (node or link) to groups of nodes to the whole system. Moreover, networks can support dynamical processes of various types, from spreading to synchronization phenomena[3]. Thus, understanding how network features impact such processes, or which parts of a network play the most important role, is of crucial relevance. For instance, hubs, nodes with a very large number of connections (degree), are known to influence processes, such as spreading or opinion dynamics, because of their tendency to be reached easily, and of their ability to transmit to many other nodes[1,3]. The statistics of the individual number of connections of nodes are, however, not a sufficiently rich characterization: the existence of well-connected groups of nodes might be even more relevant. For instance, the tendency of hubs to be connected to each other far above chance is quantified by the rich-club coefficient[5]. A more systematic way to decompose a network into a hierarchy of subgraphs of increasing connectedness is given by the $k$-core decomposition[6-9]: the $k$-core of a network is the maximal

subgraph such that all its nodes have a degree (number of neighbours in the subgraph) at least $k$. This decomposition provides a fingerprint of the network's structure[8,10-12], gradually focusing on more densely interconnected parts of the network that were shown to play a crucial role in spreading processes[13-15]. In fact, the coreness of a node, defined as the largest value of $k$ such that the node belongs to the corresponding $k$-core, gives a centrality measure that largely determines the impact of a spreading process initiated (seeded) in that node[13]. This decomposition has also been extended to weighted networks[16], via the $s$-core decomposition (where $s$ represents the strength of a node, i.e., the sum of the weights of its adjacent links)[17], to temporally evolving networks[18,19], to multilayer networks[20] and to bipartite networks[21-23].

Despite their convenience, network representations are limited to systems composed of only dyadic interactions. However, recent works have made clear that many real systems include interactions between groups of units[24,25]. Examples range from group conversations[26] to research teams[27], from neural systems[28] to interactions between species in ecosystems[29]. Analogously, considering a purely dyadic network substrate for the unfolding of processes, such as consensus formation or (social) contagion, could put a limit on the ability to describe key

[1]Aix Marseille Univ, Université de Toulon, CNRS, CPT, Turing Center for Living Systems, Marseille, France. [2]Network Science Institute, Northeastern University London, London E1W 1LP, United Kingdom. [3]Department of Network and Data Science, Central European University, 1100 Vienna, Austria. [4]CENTAI, Corso Inghilterra 3, 10138 Turin, Italy. ✉e-mail: alain.barrat@cpt.univ-mrs.fr

mechanisms that are at play. For instance, reinforcement mechanisms –in which two or more people can convince others in a group conversation–cannot be naturally accounted for by considering only dyadic interactions[30–33]. In these cases, systems and processes can be effectively represented within the framework of hypergraphs, a "higher-order" generalization of networks in which nodes can interact in hyperedges, groups of arbitrary size[25,34,35]. Higher-order interactions give rise to both novel structures[36–38] and phenomena[24,39], highlighting the importance of characterization tools able to detect hierarchies and relevant subparts of systems that are better represented by hypergraphs.

Here, we contribute to this endeavour by studying the decomposition of a hypergraph in $(k, m)$-hyper-cores, which are defined as a series of subhypergraphs of increasing connectivity $k$, ensured by hyperedges of increasing sizes $m$[40] (this definition is, in fact, equivalent to the one of two-mode cores in bipartite networks[21–23]). We apply this decomposition to a wide range of data sets, representing systems of different nature: this highlights how such decomposition identifies non-trivial mesoscopic higher-order structures, in particular when comparing it to the one obtained in suitable null models. The decomposition in hyper-cores leads us to the definition of the hyper-coreness, a new family of centrality measures for nodes in hypergraphs based on their degree of inclusion in hyper-cores. Finally, we investigate the role of the hyper-cores, and of the nodes with the largest hypercoreness, in paradigmatic spreading and consensus processes based on group interactions[32,41,42]. We show that spreading processes tend to be localized on hyper-cores associated to large $k$ and $m$. We then study the performance of hypercoreness-based strategies to identify influential nodes in sustaining and driving higher-order processes. We find that hypercoreness can be effectively used to maximise the total outbreak size in higher-order spreading processes[41,42] and to help committed minorities reach the tipping point leading to the systemic takeover in social convention games[43].

## Results

### Hyper-core decomposition and hypercoreness

The hyper-cores, i.e. the higher-order cores of a hypergraph, allow us to define a systematic decomposition of a hypergraph in a double hierarchy of nested subhypergraphs of increasing connectedness and hyperedge sizes. Let us consider a (static) hypergraph $\mathcal{H} = (\mathcal{V}, \mathcal{E})$, where $\mathcal{V}$ is the set of its $N = |\mathcal{V}|$ nodes and $\mathcal{E}$ is the set of its hyperedges[25]. We recall that a hyperedge $e = \{i_1, i_2, \ldots, i_m\}$ is a set of $m$ nodes, which can thus represent a group interaction between these nodes. We denote by $M = \max_{e \in \mathcal{E}} |e|$ the largest hyperedge size in $\mathcal{H}$. Each node $i \in \mathcal{V}$ can be characterized by a vector of degrees $\boldsymbol{d}(i) = [d_2(i), d_3(i), \ldots, d_m(i), \ldots, d_M(i)]$ whose component $d_m(i)$ denotes the $m$-hyper-degree of the node $i$, i.e., the number of distinct hyperedges of size $m$ to which it belongs. We denote by $D_m(i) = \sum_{p \geq m} d_p(i)$ the number of distinct hyperedges of size at least $m$ to which $i$ belongs.

The $(k, m)$-hypercore is defined as the maximum subhypergraph $\mathcal{J}$ induced by the set of nodes $\mathcal{A} \subseteq \mathcal{V}$ and with hyperedges of size at least $m$, such that $\forall i \in \mathcal{A}, D_m^{\mathcal{J}}(i) \geq k$, where $D_m^{\mathcal{J}}(i)$ denotes the number of distinct hyperedges of size at least $m$ in which $i$ is involved within the subhypergraph $\mathcal{J}$[40]. In other terms, all the nodes in the $(k, m)$-hyper-core belong to at least $k$ hyperedges of size at least $m$, within the hyper-core itself. The set of hyperedges of the subhypergraph $\mathcal{J}$, induced by the set $\mathcal{A} \subseteq \mathcal{V}$, is defined by $\mathcal{S} = \{e \cap \mathcal{A} \text{ s.t. } e \in \mathcal{E} \wedge |e \cap \mathcal{A}| \geq m\}$[44], i.e., a hyperedge of $\mathcal{S}$ is a subset of a hyperedge of $\mathcal{E}$, of size at least $m$ and containing only nodes of $\mathcal{A}$. Note that hyperedges of $\mathcal{S}$ might thus not be in $\mathcal{E}$, but they can still be interpreted as existing interactions if one assumes that subsets of a set of interacting nodes are indeed interacting. As our study will focus on the sets of nodes forming the various hyper-cores, rather than on their sets of hyperedges, this consideration does not impact our results. We also note that this definition of hyper-cores is equivalent to the one of two-mode cores in bipartite

networks, upon mapping a hypergraph onto a bipartite representation, in which nodes represent either hyperedges or nodes of the hypergraph, and each hyperedge is connected to its elements[21–23,45]. The $(k, m)$ two-mode-core of a bipartite graph corresponds indeed to the bipartite subgraphs in which the nodes have degree respectively at least $m$ (for the nodes representing hyperedges) and $k$ (for the nodes representing nodes of the hypergraph). The earlier works introducing such concepts[21–23] have indeed mostly focused on their interpretation in bipartite networks, rather than for hypergraphs (see however[40]), and have shown their interest for visualisation purposes[21] but did not study how empirical data can be systematically decomposed into hyper-cores, nor the interplay between hyper-cores and dynamical processes on hypergraphs.

To obtain the $(k, m)$-hyper-core of a hypergraph, one can first remove from $\mathcal{E}$ all hyperedges of size smaller than $m$. One then removes recursively from $\mathcal{V}$ all nodes $i$ with $D_m(i) < k$, until all the nodes in the remaining subhypergraph are involved in at least $k$ hyperedges of size at least $m$. Note that this process does not correspond only to the removal of nodes with $D_m(i) < k$ in the original hypergraph $\mathcal{H}$: indeed, each time a node is removed, the sizes of the hyperedges to which it belongs decrease by one unit. Thus, the removal of a node can induce the removal of some of the hyperedges to which it belongs, if their size becomes less than $m$, or if they fully coincide with already existing hyperedges. In Fig. 1 we illustrate the process on an example hypergraph and highlight some of its $(k, m)$-hyper-cores. The straightforward implementation of the procedure to obtain the complete $(k, m)$-core structure of a hypergraph $\mathcal{H} = (\mathcal{V}, \mathcal{E})$ features a time complexity that scales as $M(N + |\mathcal{E}| \log(|\mathcal{E}|))$ (see the Code Availability for an implementation, and the Supplementary Note 7 in

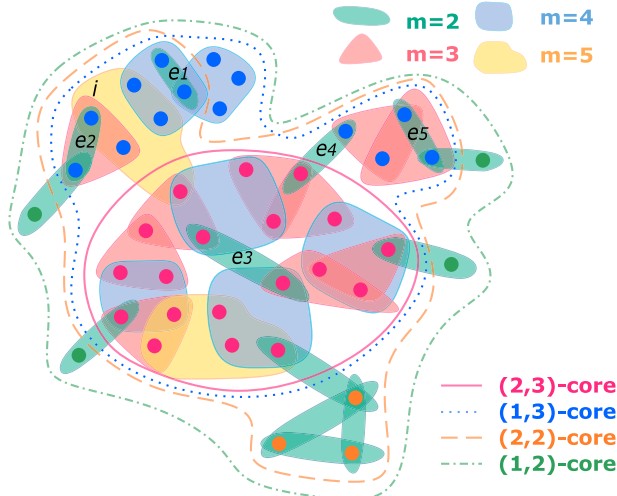

**Fig. 1 | Sketch of the $(k, m)$-hyper-core decomposition.** We show a hypergraph and highlight some of its $(k, m)$-hyper-cores. Note the inclusions as $k$ or $m$ increase: the (1, 2)-hyper-core contains the (1, 3)-hyper-core, which contains the (2, 3)-hyper-core; similarly the (1, 2)-hyper-core contains the (2, 2)-hyper-core which contains the (2, 3)-hyper-core. On the other hand, the (1, 3)-hyper-core and the (2, 2)-hyper-core share some nodes but neither is included in the other. The green nodes belong to the (1, 2)-hyper-core but neither to the (1, 3)- nor the (2, 2)- ones. The blue nodes belong to the (1, 3)-hyper-core but are excluded from the (2, 3) one. Orange nodes belong to the (2, 2)-hyper-core but are excluded from the (2, 3) one because they belong only to hyperedges of size 2. The (1, 4)-core and (1, 5)-core contain all the nodes involved respectively in at least one interaction with $m \geq 4$ and $m \geq 5$ (for simplicity these cores are not highlighted). The $(k, 2)$-cores and $(k, 3)$-cores with $k \geq 3$, and the $(k, 4)$-cores and $(k, 5)$-cores with $k \geq 2$ are all empty. Notice that the node $i$ does not belong to the (2, 3)-core even if $D_3(i) = 2$ because of the recursive and interaction downgrading mechanisms of the decomposition; in the (1, 3)-core and (2, 3)-core the pairwise interactions $e_i \forall i \in [1, 5]$ are excluded, thus the (1, 3)-core is composed of two disjoint subhypergraphs.

the Supplementary Information, SI, for further details; we also note that efficient algorithms have been proposed in the context of bipartite graphs[23]).

As $k$ and $m$ increase, the $(k, m)$-hyper-cores progressively identify groups of nodes increasingly connected with each other through interactions of increasing order. In fact, the $(k, m)$-hyper-core includes the $(k, m+1)$- and $(k+1, m)$-hyper-cores (Fig. 1). We define the $m$-shell index $C_m(i)$ of a node $i$ as the value of $k$ such that $i$ belongs to the $(k, m)$-hyper-core but not to the $(k+1, m)$-hyper-core. The $(k, m)$-shell $S_{(k, m)}$ can then be defined as the set of all nodes whose shell index $C_m(i)$ at size $m$ is $k$, and we denote by $k_{max}^m$ the maximum value of $k$ such that the shell $S_{(k, m)}$ is not empty. The ratio $C_m(i)/k_{max}^m$ thus quantifies how well-connected node $i$ is in the hypergraph when considering group sizes at least $m$. As this ratio is a function of $m$, different nodes have different functions, which can potentially exhibit very different functional shapes (see Supplementary Figure 5 in the SI for examples). It is therefore impossible to use such functions to compare and rank nodes. This suggests to use another strategy, namely, to construct a scalar capable of summarizing the centrality properties of a node with respect to the hyper-core decomposition. We thus define a family of centrality measures that we call hypercoreness. Specifically, we define for each node $i$ its g-hypercoreness $R_g(i)$ as:

$$R_g(i) = \sum_{m=2}^{M} g(m) C_m(i)/k_{max}^m, \qquad (1)$$

where $g(m)$ is an arbitrary weight function, which can weigh differently the various possible sizes of higher-order interactions. $R_g$ is thus now a scalar that can be used to rank nodes. The simplest case is given by the *size-independent hypercoreness R*, which weighs equally all group sizes by using $g(m) = 1, \forall m$. Alternatively, the function $g$ could be used to emphasise hyperedges of larger or smaller sizes, or a specific value (e.g. by using $g(m) = \delta(m - m^*)$ if $m^*$ is a specific size of interest for a dynamical process). In the spirit of a data-driven measure, we also consider the *frequency-based hypercoreness R_w*, where the function $g$ is informed by each data set and weighs each group size $m$ by its relative abundance in the data:

$$R_w(i) = \sum_{m=2}^{M} \Psi(m) C_m(i)/k_{max}^m, \qquad (2)$$

where $\Psi(m)$ is the fraction of hyperedges of size $m$ in the considered data set. The rationale behind using such a weight function is to give more importance to the more frequent hyperedge sizes.

## Hyper-core decomposition of empirical hypergraphs

To illustrate the decomposition processes along $(k, m)$-hyper-cores, we rely on a number of empirical hypergraphs, obtained from publicly available data sets, that describe a variety of systems of agents interacting in different environments. In particular, we consider data sets of face-to-face interactions between individuals, collected in contexts ranging from workplaces to schools[46–49]. We also use data sets of email communication (email-EU, email-Enron[50–52]) and of other online interactions: online reviews of products (music-review[52,53]) or opinion exchanges in scientific forums[52,54]. We moreover consider data describing committees membership (house-committees, senate-committees[52,55,56]) and bills sponsorship (congress-bills, senate-bills[52,55,57,58]) in the US Congress. Finally, we use ecological data sets, describing pollination interactions between plants and insects species[59–61]. These data sets cover a wide range of system sizes and of interaction size distributions (see Methods and Supplementary Note 1 in the SI, for a detailed description of each data set). In the following, we give results on the music-review, email-EU, house-committees, and congress-bills data sets, and we refer to the SI for the other data sets.

Figure 2 shows the results of the hyper-core decomposition on two data sets. The relative size $n_{(k, m)}$ of the $(k, m)$-hyper-cores exhibits distinct behaviors as a function of $k$ and $m$, identifying structural differences between data. In some cases, the decrease with $k$ is rather smooth (Fig. 2a and Supplementary Figs. 2 and 3 in the SI), showing that most shells are populated. In other cases, abrupt drops and plateaus can be observed (Fig. 2e and Supplementary Figs. 2 and 3 in the SI), corresponding to alternatively empty and densely populated $(k, m)$-shells (see also Supplementary Figure 4 in the SI for the sizes of the $(k, m)$-shells vs. $k$ and $m$).

These differences indicate that the $(k, m)$-hyper-cores could be used to provide a fingerprint of hypergraphs, just as the $k$-core decomposition provides a fingerprint of networks[8,10,12]. We explore this point further in Fig. 2b, f, by comparing the hyper-core decomposition of empirical data with the ones of randomized versions that preserve the distribution of hyperedges sizes and the hyper-degrees of each node (see Methods for details on the randomization). The most common pattern obtained in the data sets considered (see also SI) consists in significantly smaller hyper-core sizes in the data for low values of $m$ and $k$, and significantly larger sizes at large values of $m$ and $k$. In particular, $k_{max}^m$ is most often smaller in the data for $m \leq m_0$ ($m_0 = 3$ in Fig. 2b) but larger for $m > m_0$ (see also SI). This shows the existence of structures that are more strongly connected by hyperedges of large size in the data than in their randomized counterparts, i.e., that cannot be explained by the distribution of hyperedge sizes nor by the heterogeneity of node degrees. Such results provide evidence of non-trivial hierarchical arrangement of hyperedges connectivity in data, which should thus be taken into account for realistic hypergraph modeling.

The distributions of hypercoreness values also differ across data sets, as illustrated in the rank-order plots of Fig. 2c, d, g, h and in the SI: some data sets have an almost uniform distribution of values, others feature few nodes with high hypercoreness and many nodes with medium hypercoreness, or vice-versa. We also show in the SI some typical examples of the normalized $m$-shell index function $C_m(i)$ as a function of $m$ for various nodes. As anticipated above, the diversity of these functions and of their shapes makes it difficult to compare them and justifies the need to define summary indices such as the hypercoreness.

We finally compare in the insets of Fig. 2c,d,g,h the hypercoreness $R$ and $R_w$ with the centrality of nodes obtained by disregarding the higher-order nature of the interactions and projecting the hypergraph $\mathcal{H}$ onto a network. To this aim, we transform each hyperedge in a network clique, and each edge $(i, j)$ of the resulting network is weighted by the number of distinct hyperedges in $\mathcal{H}$ involving both $i$ and $j$. We then perform the $s$-core decomposition of this weighted network and assign its $s$-coreness $S(i)$ to each node $i$[17]. As expected, since all measures deal with coreness concepts, $S(i)$ and $R(i)$ are positively correlated, as well as $S(i)$ and $R_w(i)$. However, they do not provide exactly the same information, and the hypercoreness measures enhance the information given by the $s$-coreness by providing an internal hierarchy within the nodes of maximal $s$-coreness, thanks to the fact that the hypercoreness centralities take into account not only the connectivity but also the sizes of the connecting hyperedges. That is, nodes presenting the same $s$-coreness values can span a broad range of hypercoreness values.

Having illustrated the relevance of the hyper-cores on empirical hypergraphs, we now move to study the role that these substructures play in dynamical processes on hypergraphs. In particular, we are going to investigate whether the $(k, m)$-hyper-cores and the hypercoreness centralities can be used to identify nodes and structures relevant for spreading and consensus processes whose mechanisms are explicitly defined on hyperedges. To this aim, we will consider different models of spreading processes that have been recently well studied and shown to exhibit interesting new phenomenology driven

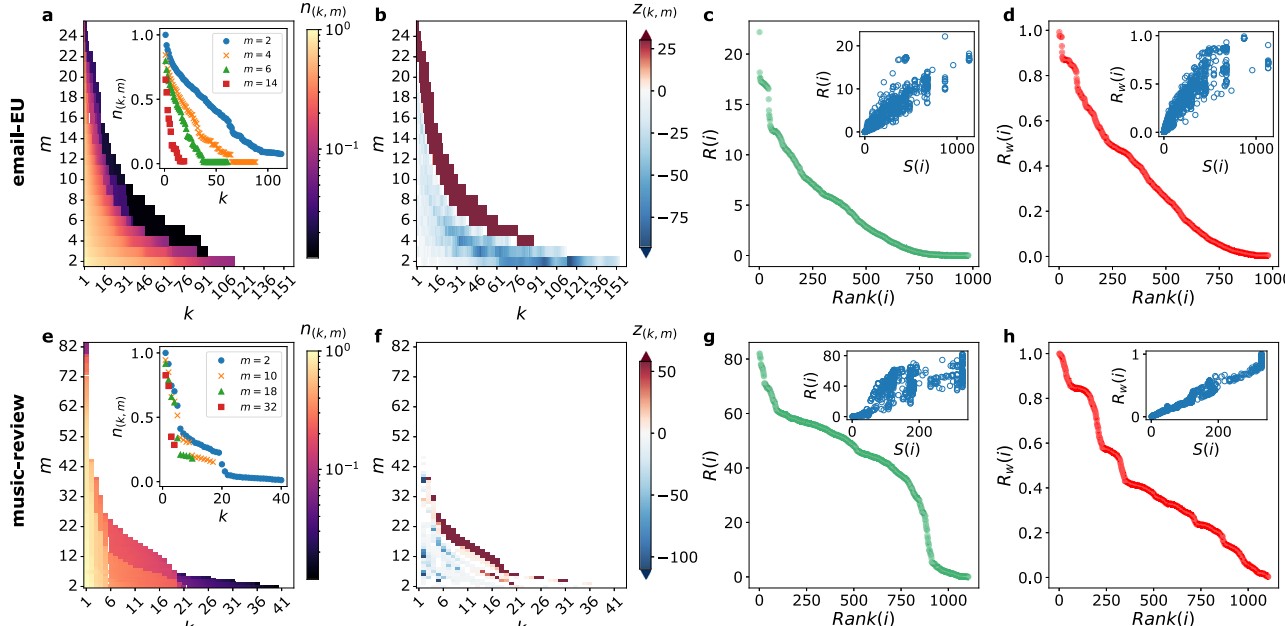

**Fig. 2 | Hyper-core decomposition of empirical hypergraphs.** Panels **a**, **e** show colormaps giving the relative size $n_{(k,m)}$ (number of nodes in the hyper-core, divided by the total number of nodes $N$) of the $(k,m)$-hyper-core as a function of $k$ and $m$ (white regions correspond to $n_{(k,m)} = 0$). In the insets, $n_{(k,m)}$ is shown as a function of $k$ at fixed values of $m$. Panels **b**, **f** show colormaps giving the z-score $z_{(k,m)}$ of the $(k,m)$-hyper-core relative size, with respect to $10^3$ shuffled realizations of the hypergraph, as a function of $k$ and $m$ (values of $z_{(k,m)} \in (-1.96, 1.96)$ are shown in white). In panels **c**, **g** the size-independent hypercoreness $R(i)$ is plotted as a function of the corresponding node rank; the insets give scatterplots of $R(i)$ vs. the $s$-coreness, $S(i)$, for all nodes. Panels **d**, **h** are the same as **c**, **g**, but for the frequency-based hypercoreness $R_w(i)$. In panels **a**–**d** we consider the email-EU data set: $R(i)$ and $S(i)$ have a Pearson correlation coefficient of $\rho = 0.90$ ($p$-value $p \ll 0.001$) and the corresponding rankings have a Kendall's $\tau$ coefficient of $\tau = 0.85$ ($p \ll 0.001$), while $R_w(i)$ and $S(i)$ have $\rho = 0.90$ ($p \ll 0.001$) and $\tau = 0.85$ ($p \ll 0.001$); in panels **e**–**h** we consider the music-review data set: $R(i)$ and $S(i)$ have $\rho = 0.74$ ($p \ll 0.001$) and $\tau = 0.58$ ($p \ll 0.001$), while $R_w(i)$ and $S(i)$ have $\rho = 0.98$ ($p \ll 0.001$) and $\tau = 0.89$ ($p \ll 0.001$).

by higher-order effects[41,42], and a consensus formation model that has been shown to reproduce well-experimental results on the effect of critical masses of committed individuals[62], and where higher-order effects have also been shown recently to influence this phenomenology[32].

## Higher-order contagion processes localize in hyper-cores, and high hypercoreness seeds increase total outbreak size

Networks are widely used to describe the substrate on which contagion processes take place, such as the spread of pathogens or information. In standard diffusion modeling approaches, nodes represent individuals that at any time can be in one of several possible states, such as $S$ (susceptible), $I$ (infectious) or $R$ (recovered); $S$ nodes become $I$ at rate $\beta$ when they share a link with an infectious ($I$) individual, while infected ($I$) nodes recover spontaneously at rate $\mu$, either becoming again susceptible ($S$), in what is usually called the SIS model[63], or becoming recovered ($R$) in the so-called SIR model. Recently, several models have been proposed to take into account possible higher-order mechanisms, that amount to reinforcement mechanisms affecting the contagion probability due to the simultaneous exposure to multiple sources of infections in group interactions[30,41,64,65]. For instance, in a social contagion process, the probability that an individual is convinced upon separate exposures to two "infectious" neighbours can be reinforced if these exposures occur during a group discussion featuring the three individuals altogether.

Here, we show that hyper-cores and nodes with large hypercoreness centralities play a crucial role in the dynamics of higher-order spreading processes. To this aim, we consider the recently proposed higher-order nonlinear contagion[41]. In this model, each susceptible node in a hyperedge of size $m$ in which there are $i$ infected individuals becomes infectious with rate $\lambda i^\nu$, where $\nu$ controls the non-linearity of the process (for $\nu = 1$ the usual linear contagion is recovered, while for

$\nu > 1$ non-linearities are introduced) and $\lambda \in [0, 1]$ (see Methods for details). Infected individuals ($I$) recover independently at constant rate $\mu$, becoming either susceptible $S$ (SIS model) or $R$ (SIR). The higher-order nature of contagion produces novel effects on the epidemic phenomenology, including abrupt transitions with bistability in the SIS phase diagram and intermittent regimes[42,64]. Moreover, hyperedge size has been shown to play an important role for such higher-order nonlinear contagion processes: on the one hand, in a stationary state, the infection tends to localize on large hyperedges[41]; on the other hand, nodes belonging to large groups are optimal seeds—in terms of spreading speed—at the beginning of an outbreak[41]. Nevertheless, which nodes among these large groups are most important for the contagion, both in terms of being infectious more often in an SIS process, or in terms of having large spreading power, remains an unexplored issue. In spreading processes on networks the coreness has been shown to correlate with spreading properties of nodes[13]. Thus, here it seems natural to investigate which role the connectivity properties of large hyperedges play in higher-order contagion processes: does the infection process localize more strongly in hyper-cores of large $k$ and $m$ and/or on nodes with large hypercoreness values? Do nodes with higher hypercoreness have larger spreading power?

To investigate these points, we perform numerical simulations of the higher-order nonlinear contagion model on empirical hypergraphs. In the SIS case, the system is initialized with one single seed of infection (a randomly chosen node in state $I$) in an otherwise fully susceptible population. We let the process evolve (see Methods) until a steady state is reached in which the number of infectious individuals fluctuates (we consider parameter values such that the epidemic does not die out rapidly). We then consider a finite time-window $T$ and measure for each node $j$ the time $\tau(j)$ it spends in the $I$ state during that window. In this way we identify the nodes on which the epidemic is

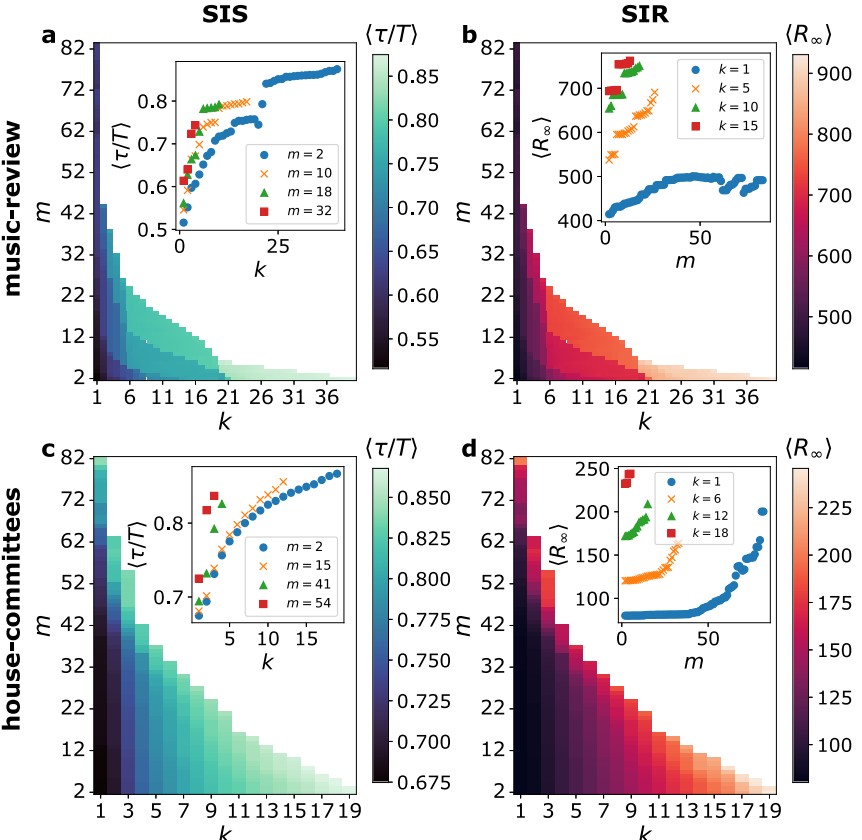

**Fig. 3 | Hyper-cores for seeding and localization in higher-order nonlinear contagion processes.** For the SIS model, panels **a** and **c** give the heatmap of the average fraction of time $\langle \tau/T \rangle$ of infected nodes in the steady state as a function of $k$ and $m$. Averages are computed over all the nodes of each $(k,m)$-hyper-core. The insets represent $\langle \tau/T \rangle$ as a function of $k$ for fixed values of $m$. All results are obtained by averaging the results of $10^3$ numerical simulations, with an observation window $T = 10^3$. For the SIR model, panels **b** and **d** show the heatmap of the average final size of the epidemic $\langle R_\infty \rangle$ as a function of $k$ and $m$, where the process is seeded in a single node belonging to the $(k,m)$-hyper-core (averaged over all nodes of the hyper-core). The insets represent $\langle R_\infty \rangle$ as a function of $m$ for fixed values of $k$. All results are obtained by averaging the results of 300 numerical simulations for each seed. Panels **a** and **b**: music-review data set with $\nu = 1.25$, $\lambda = 5 \times 10^{-4}$ (**a**) and $\nu = 3$, $\lambda = 5 \times 10^{-4}$ (**b**). Panels **c** and **d**: house-committees data set with $\nu = 1.25$, $\lambda = 5 \times 10^{-4}$ (**c**) and $\nu = 4$, $\lambda = 5 \times 10^{-5}$ (**d**). In all panels $\mu = 0.1$.

mainly localized in the steady state, i.e. the nodes that drive and sustain the process. In the SIR case instead, the dynamics starts from a single seed and evolves until no individual is in the state $I$ anymore (only nodes in states $S$ or $R$ remain). To quantify the "spreading power" of each node $j$ considered as an individual seed, we average the final epidemic size $R_\infty(j)$, i.e., the number of nodes in state $R$ at the end of the process, over 300 stochastic runs for each seed.

Figure 3 reports results of simulations performed on the music-review and house-committees data sets (see SI for the other data sets). Panels 3a and c show that nodes in $(k,m)$-hyper-cores with either increasing $k$ or $m$ tend to be more often infectious during the SIS process, as $\tau(j)/T$ averaged over all nodes of each $(k,m)$-hyper-core increase with $k$ and $m$. This implies that the SIS process is more localized in the $(k,m)$-hyper-cores with large $k$ (which favors connectedness, hence mutual reachability) and $m$ (i.e., large hyperedges where large values of $i$ can be obtained yielding large infection rates). The insets show how the dependency on the connectivity $k$ is non-trivially affected by $m$, the minimal group size considered. Moreover, Fig. 3b and d show that the final epidemic size $\langle R_\infty(j) \rangle$ of SIR processes, averaged over all nodes of each $(k,m)$-hyper-core, increases both with $k$ and $m$, with the insets emphasizing how the minimal connectivity $k$ impacts the dependency on the group-size $m$.

Many centrality measures have been defined for nodes in a network. Among them, the coreness centrality is particularly suited to identify important nodes in spreading processes on networks[13]. Moreover, it has been shown that nodes belonging to hyperedges of large size are important in nonlinear spreading on hypergraphs[41]. These earlier results, together with the results of Fig. 3, prompt us to investigate whether the hypercoreness centrality measures are able to identify the nodes with the most important role in the higher-order nonlinear contagion process, and to compare their performance with coreness concepts based on a network representation that does not take group sizes explicitly into account. We thus rank the nodes according to the fraction of time $\tau/T$ spent in the I state during the SIS process. Figure 4a and c show the Jaccard coefficient between the first $fN$ nodes according to this ranking and the first $fN$ nodes according to a ranking based on one of the considered centralities: the size-independent hypercoreness $R$, the frequency-based hypercoreness $R_w$, the $s$-coreness $S$, and the $k$-coreness (unweighted version of the $s$-coreness). The larger the Jaccard coefficient is, the better the centrality identifies nodes on which the SIS process tends to be localized. The results show that both $R$ and $R_w$ are more able to uncover the 10% of nodes where the process is most localized, with especially good performances obtained by the frequency-based hypercoreness in some data sets. The insets of panels a and c present similar results under a different angle: namely, they display the average of $\tau/T$ over the $fN$ nodes with the highest hypercoreness $R$ or $R_w$, or the highest $k$- or $s$-coreness. Nodes with highest coreness tend to be more often in the infectious state, and this tendency is stronger for the hypercoreness centralities than for the $k$ and $s$-coreness: among the nodes with the largest values of $k$- or $s$-coreness, the hypercoreness centralities allow to distinguish which ones are the most involved in the higher-order spreading processes. Overall, the hypercoreness centralities thus

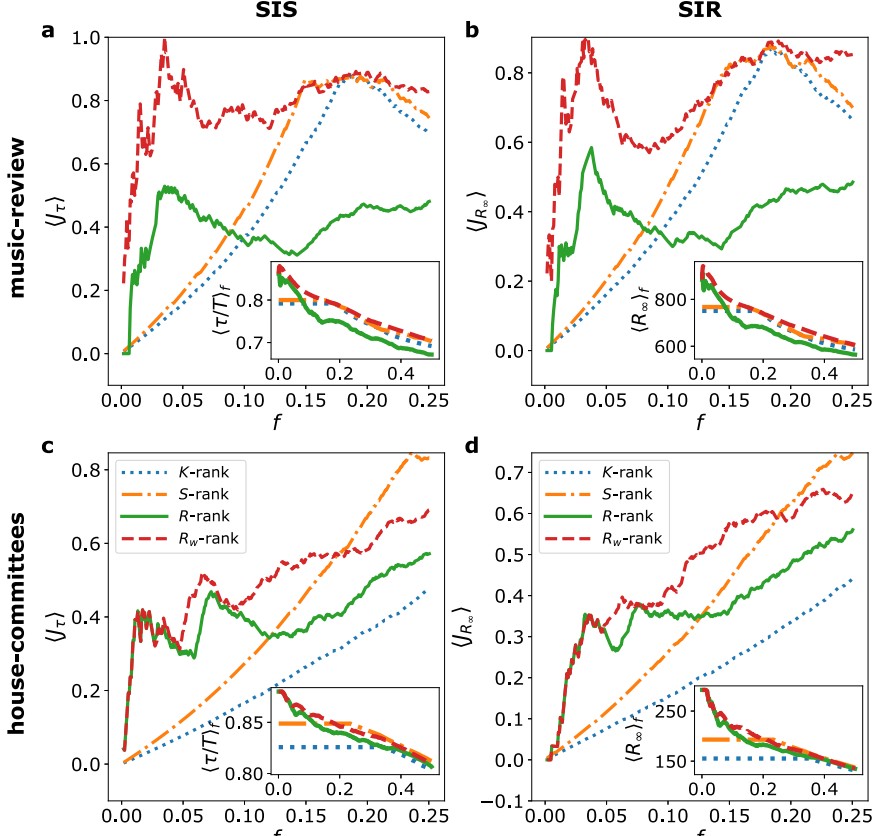

**Fig. 4 | Centralities performance in identifying nodes with highest importance in higher-order nonlinear contagion processes.** Panels **a**, **c** give the average Jaccard similarity $\langle J_\tau \rangle$ between the nodes in the top $fN$ positions of the rankings based either on the fraction of time $\tau/T$ spent in the I state during the SIS process, or on each of the centralities considered (see legend), vs. $f$. The insets represent, as a function of $f$, the fraction $\langle \tau/T \rangle_f$ averaged over the first $fN$ nodes according to the different coreness rankings. Panels **b**, **d** show the average Jaccard similarity $\langle J_{R_\infty} \rangle$ between the nodes in the top $fN$ positions of the rankings based either on $R_\infty$, i.e. the average epidemic final-size produced by seeding the SIR process in each node, and each of the centralities considered, vs. $f$. The insets give the average epidemic final-size $\langle R_\infty \rangle_f$, averaged over the first $fN$ nodes according to coreness rankings, as a function of $f$. Panels **a**, **b** refer to the music-review data set, panels **c**, **d** refer to the house-committees data set. The parameters and simulation conditions are fixed as in Fig. 3.

perform better at identifying nodes on which the spreading gets more localized than coreness measures that ignore the size of hyperedges, i.e., are based on a network representation.

The panels **b** and **d** of Fig. 4 convey similar results for the SIR case: hypercoreness centralities better identify the nodes with highest spreading power than coreness centralities which do not take group sizes into account, and the nodes with higher hypercoreness lead to larger epidemics (insets), determining a hierarchy even among the nodes with the highest $k$- or $s$-coreness; nodes with higher connectedness along groups of larger sizes can seed more efficiently the contagion process, and the hypercoreness centralities identify well the nodes with the highest spreading power.

In the SI we show that a similar phenomenology is obtained with a different model of contagion involving higher-order mechanisms[42,64], for both SIS and SIR.

### Hyper-core seeding facilitates systemic takeover by minority norms

Group interactions can also play an important role in the formation of consensus and the emergence of shared conventions in a population. In the context of addressing societal challenges, critical mass theory predicts that regular individuals might benefit from the presence of a committed minority that aims at overturning the status quo[66]. Recently, it has been shown that group interactions can influence this takeover[32]. An important issue in this respect concerns the best "seeding" strategy –where should the committed minority start from

in order to best achieve the takeover? Here we show how hypercoreness centralities can provide an answer.

We consider the well-known naming-game (NG) model[43], which describes how a shared convention can emerge in a population of interacting agents[62,67,68], in its minimal version modified to take group interactions into account[32]. Individuals are represented by the $N$ nodes of a hypergraph, and each node is endowed with a dictionary that can contain at most two names (representing conventions or norms), $A$ and $B$. At each time-step a hyperedge is chosen randomly and a speaker is randomly selected within it. The speaker randomly chooses a name from its dictionary and communicates it to the other hyperedge members (the listeners), who can agree or not on the proposed name. To determine the possibility of an agreement within the hyperedge, we consider two alternatives[32]: (*i*) the union rule, for which an agreement can be reached if at least one of the listeners has the proposed name in its dictionary; (*ii*) the unanimity rule, for which the agreement can be reached only if all nodes in the group have the proposed name in their dictionary. A parameter $\beta \in [0, 1]$ modulates the social influence by controlling the propensity of the listeners to accept the local consensus: the group agreement becomes effective only with probability $\beta$. In this case, all nodes in the hyperedge add the accepted name to their dictionary, if it was not already present, deleting all others. If instead no agreement is reached, the listeners simply add the name given by the speaker to their dictionaries.

The population includes a committed minority of $N_p$ individuals who do not obey these rules whenever they are listeners, but instead

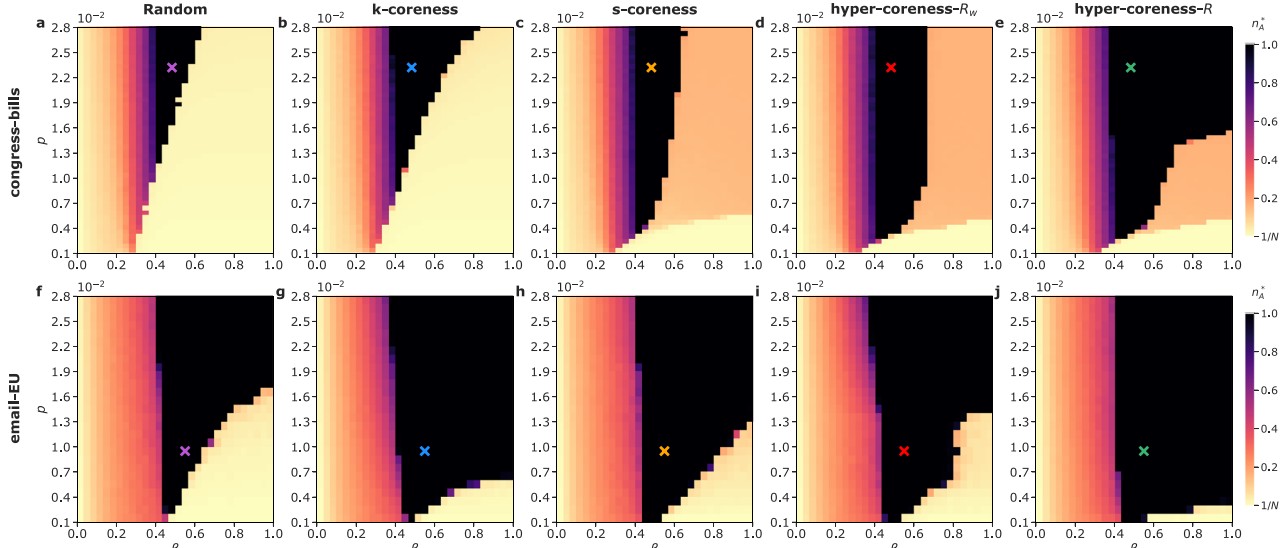

**Fig. 5 | Comparison of seeding strategies for committed minorities in a naming-game process.** The stationary fraction $n_A^*$ of nodes supporting only the name $A$ is shown as a function of the fraction of committed nodes $p$ and the agreement probability $\beta$. **a–e**: congress-bills data set with unanimity rule. **f–j**: email-EU data set with union rule. Committed nodes are selected through random seeding (**a, f**), top $k$-coreness (**b, g**), top $s$-coreness (**c, h**), top frequency-based $R_w$ hypercoreness (**d, i**) and top size-independent $R$ hypercoreness (**e, j**) strategies. With the top $R$ hyper-coreness strategy, a fraction $p = 1.51 \times 10^{-2}$ in the congress-bills data set with unanimity rule is enough to allow the minority takeover over a range of $\beta$ values whose extension is $\Delta\beta \gtrsim 0.5$. This cannot be achieved with the other strategies, for which below $p = 2.8 \times 10^{-2}$ only $\Delta\beta \sim 0.4$ can be reached (see panels **a–e**). In the email-EU data set with the union rule, a fraction $p = 4.1 \times 10^{-3}$ is enough to obtain the

minority dominance over $\Delta\beta \gtrsim 0.5$ when seeded according to the top size-independent $R$ hypercoreness strategy. With the top $s$-coreness and the random strategies the same result is obtained only for $p = 1.33 \times 10^{-2}$ and $p = 1.74 \times 10^{-2}$ respectively (panels **f–j**). The minority takeover, i.e. $n_A^* = 1$, takes place for 7.9% of the explored parameter space in panel **a**, 13.8% in **b**, 16.3% in **c**, 23.0% in **d**, 41.5% in **e**, 37.0% in panel **f**, 51.9% in **g**, 45.9% in **h**, 45.2% in **i** and 56.4% in **j**. All simulations are run until the absorbing state $n_A^* = 1$ is reached or the dynamics has evolved for $t_{max} = 5 \times 10^5$ time steps. The stationary fraction $n_A^*$ is obtained by averaging over 100 values sampled in the last $T = 5 \times 10^4$ time-steps. Results refer to the median values obtained over 200 simulations for each pair of parameter values. Cross markers indicate the $(\beta, p)$ values considered for Fig. 6.

stick to their norm, a single name $A$ (their dictionary is never updated). Such individuals have also been called "zealots" in various models of opinion dynamics[69–71]. We initiate the process with the rest of the population, i.e. the majority, having only the name $B$. The system can evolve towards different regimes of co-existence of the two names or of dominance of one name, depending on $\beta$, on the considered rule, and on the relative size of the minority $p = N_p/N$. In particular, the committed minority can overcome the majority, with the whole population converging on $A$, for a range of intermediate values of $\beta$ and for large enough $p$. When committed individuals are chosen at random in the population, this range increases when the hypergraph contains hyperedges of larger sizes[32]. This naturally raises the question of whether the committed minority might also benefit from belonging to specific substructures, such as hyper-cores with large connected-ness and group sizes.

We investigate this issue through numerical simulations of the higher-order NG process on empirical static hypergraphs, selecting committed individuals with different seeding strategies: (*i*) at random from the entire population (random); (*ii*) as the $N_p$ ones with the highest size-independent hypercoreness $R$ or frequency-based hypercoreness $R_w$ (top hypercoreness); (*iii*) as the $N_p$ ones with the highest $s$-coreness (top $s$-coreness) or $k$-coreness (top $k$-coreness) in the projected graph. In each case, we measure the fraction $n_A$ of nodes holding only $A$ in their dictionary (both committed or not), and focus on its large time limit $n_A^*$. Figure 5 reports the simulation results for two empirical data sets, congress-bills (**a–e**) and the email-EU (**f–j**) (see SI for the other data sets). For the random strategy, we recover the results of[32]: for low values of $\beta$, a co-existence state of $A$ and $B$ is observed; at a low fraction of committed and large $\beta$ values, the majority remains $B$. At intermediate $\beta$, the minority takes over and the whole population converges on $A$.

To go further, we consider non-random strategies, in which the committed individuals are selected according to a centrality

criterion. In particular, we consider different scenarios in which committed individuals are placed on the most central nodes according either to their $k$- or $s$-coreness, i.e., without taking group sizes into account, or according to one of the considered hyper-coreness centralities. Figure 5 shows, for two data sets, that even if different scenarios yield the same phenomenology, the choice of the seeding strategy can strongly enhance the range of parameters in which the minority overturns the majority (black-coloured regions). Results on the other data sets are reported in the SI. We note that the size-independent hypercoreness $R$ tends to be globally more effective at enabling the minority takeover than the frequency-based one $R_w$. This might be due to the fact that seeding and convincing very large groups can have an enormous effect in the NG dynamics, even if they are rare in the data (belonging to such large groups is less emphasized in $R_w$ than in $R$). In general, a tiny fraction of committed individuals, selected according to their hypercoreness centrality, is able to take over on a wide range of $\beta$ values (for low $\beta$ values, a co-existence regime is observed whatever the seeding strategy –due to the small propensity to accept a local consensus[32]). The value of critical mass $p_c$ necessary to bring the system to the tipping point at fixed $\beta$ is also strongly lowered for the top $R$ hypercoreness strategy. For instance, in the congress-bills data set with unanimity rule and $\beta = 0.62$, the critical mass for the top size-independent hypercore-ness strategy is $p_c^R = 6.4 \times 10^{-3}$ ($p_c^{R_w} = 7.0 \times 10^{-3}$ for the top frequency-based hypercoreness strategy), as compared to $p_c^r = 2.68 \times 10^{-2}$, $p_c^k = 2.2 \times 10^{-2}$ and $p_c^s = 2.04 \times 10^{-2}$ obtained with the random, the top $k$-coreness and top $s$-coreness strategies respectively (see Fig. 5a–e); similarly, in the email-EU data set with union rule and $\beta = 0.83$, these values are respectively $p_c^R = 3.1 \times 10^{-3}$, $p_c^{R_w} = 1.3 \times 10^{-2}$, $p_c^r = 1.53 \times 10^{-2}$, $p_c^k = 6.1 \times 10^{-3}$, $p_c^s = 9.2 \times 10^{-3}$ (see Fig. 5f–j).

The hypercoreness centralities are overall particularly effective in identifying nodes with a crucial role in higher-order NG processes.

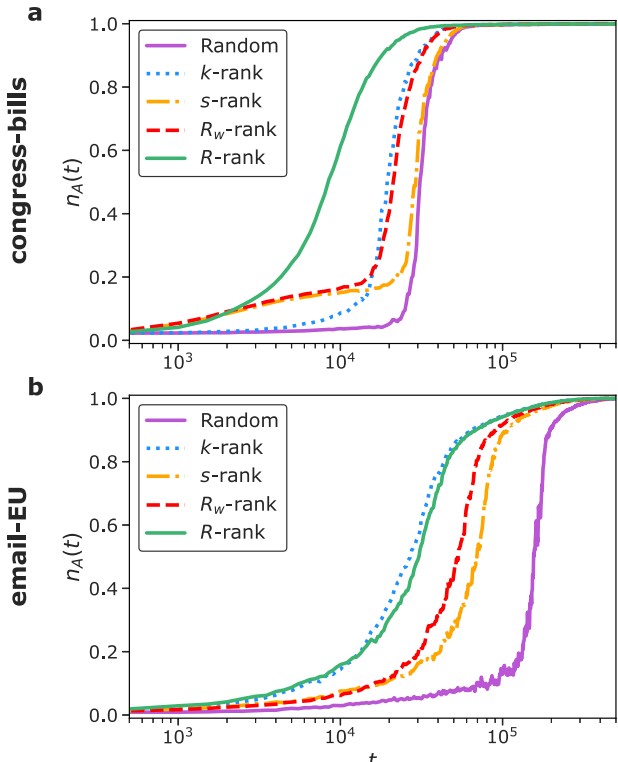

**Fig. 6 | Temporal dynamics of minority takeover with different seeding strategies for committed minorities in the naming-game process.** Panels **a**, **b** show the temporal evolution of the fraction of nodes supporting only the name $A$, $n_A(t)$, for the different seeding strategies for the committed minority and for fixed values of the agreement probability $\beta$ and of the fraction of committed nodes $p$ (see the cross markers in the heatmaps of Fig. 5), (**a**): congress-bills data set with unanimity rule and $(\beta, p) = (0.48, 2.3 \times 10^{-2})$; (**b**): email-EU data set with union rule and $(\beta, p) = (0.55, 9.2 \times 10^{-3})$. All results are obtained in the same simulation conditions of Fig. 5.

Indeed, nodes belonging to $(k, m)$-hyper-cores with large values of $k$ and $m$, if committed, can convince many others through their simultaneous presence in several large groups. This is efficiently sustained by their large connectedness, favouring convergence on their convention even outside of the committed minority. In addition, Fig. 6 illustrates how, even when all seeding strategies lead to the agreement on the convention initially supported by the minority, the hypercoreness seeding strategies lead to particularly fast convergence. As also shown for other data sets in the SI, the convergence processes obtained using seeding strategies based on hypercoreness are always among the fastest explored.

## Discussion

Here we have considered a systematic procedure to extract, from a given hypergraph, structures of increasing connectedness along increasing group sizes: the $(k, m)$-hyper-cores, in which each node is connected to the others by at least $k$ hyperedges of sizes at least $m$. We have defined a new family of centralities in hypergraphs: a node hypercoreness summarizes its relative depth in the hierarchies of hyper-cores at all orders. We have specifically considered two among the arguably most natural choices in this family, the size-independent hypercoreness, which does not put any bias towards a specific size, and the frequency-based hypercoreness, which directly takes into account the distribution of group sizes in each data set. Using empirical data describing a variety of higher-order systems, and using a comparison with a null model, we have illustrated how the $(k, m)$-hyper-cores provide a (statistically significant) fingerprint of empirical

hypergraphs. Crucially, we have also highlighted how hyper-cores with increasing $k$ and $m$ play important roles in several dynamic processes with higher-order mechanisms unfolding upon hypergraphs, such as contagion processes and consensus formation. The hypercoreness centrality identifies nodes with high spreading power and on which stationary contagion processes tend to localize; moreover nodes with high hypercoreness centralities, if belonging to a committed minority, can be particularly efficient at overturning a majority convention. As the coreness measures defined on the network representation of each data set are known to also provide indication on a node's importance for several dynamical processes, we have performed a comparison between coreness centralities that do not take into account group sizes and hypercoreness centralities. We have shown how the hypercoreness determines a hierachy among nodes with the same coreness in the projected graph, how it better identifies the most important nodes in several higher-order spreading processes and also provides powerful seeding strategies for committed individuals in the emergence of social conventions.

Our work opens the door to several research directions in the expanding field of hypergraphs structure and dynamics. It can provide an additional systematic characterization of both empirical and model hypergraphs, and thus a model validation tool as well as a comparison method between hypergraphs (e.g. by computing distances between the $(k, m)$-hypercore profiles of Fig. 2). For systems where additional properties of the nodes are known, the shell indices and hypercoreness values of nodes could be compared in more detail to provide insights into their relative positions and roles in the system. Moreover, two limitations of our study can be noted: (i) the fact that our results rely on numerical investigations, and (ii) the range of types of dynamical processes we have considered, namely spreading processes (although we considered two different higher-order infection mechanisms, and both SIS and SIR models in each case) and consensus formation. On the one hand, obtaining analytical insights on the role of various centrality measures on the spreading power of nodes in hypergraphs would be an important achievement. However, understanding which nodes are the most influential spreaders is a challenging task with very few analytical results even in usual networks (typically limited to mean-field approaches and the role of the degree centrality), while most approaches are heuristic and numerical[72–75]. On the other hand, further works should investigate the interplay between hyper-cores and hypercoreness and other dynamical processes on hypergraphs[24,33], ranging from other opinion formation models[76–78], to cooperation[79] and synchronisation[80,81]. Relevant questions could include e.g., whether nodes with higher hypercoreness can drive cooperation more efficiently, or whether synchronisation occurs preferentially, and more rapidly, in more central hyper-cores.

Moreover, while here we focused on static hypergraphs, many such systems evolve in time[82–84]. Hyper-cores and hypercoreness could be used to investigate the evolution of the higher-order interactions at multiple scales, from the global evolution of the structure described by hyper-core sizes, to the changes in shell indices and hypercoreness of individual nodes[8]. An interesting case study in this direction could be for instance the evolution of the hyper-core positions of scientists in co-authorship "networks", which are indeed evolving hypergraphs[82].

## Methods
### Data description and preprocessing
Several data sets we considered are publicly available in the form of static hypergraphs, thus they do not require any preprocessing. These data sets describe:

- email communications: within a European institution (email-EU[50]), and within Enron, between a core-set of workers (email-Enron[51,52]). Each node corresponds to an email address and a hyperedge includes the sender and all receivers of an email. Note that the original data is directed from the sender to the

receivers, but the direction is discarded when building the hyperedges.

- interactions in legislative bills in the U.S. Congress (congress-bills) and in the U.S. Senate (senate-bills)[52,55,57,58]: each node corresponds to a member of the U.S. Congress or Senate and a hyperedge involves sponsors and co-sponsors of legislative bills discussed in the Congress or Senate.

- interactions in committees in the U.S. House of Representatives (house-committees) and in the U.S. Senate (senate-committees)[52,55,56]: each node corresponds to a member of the U.S. House of Representatives or Senate and each hyperedge involves nodes that share membership in a committee.

- online interactions (3 data sets): exchanges between users of MathOverflow on algebra topics (algebra-questions) or on geometry topics (geometry-questions), in which each node corresponds to a user of MathOverflow and each hyperedge involves those users who have answered a specific question belonging to the topic of algebra or geometry[52,54]; interactions between Amazon users on music (music-review[52,53]), in which each node corresponds to an Amazon user and each hyperedge involves users who have reviewed a specific product belonging to the category of blues music.

Moreover, we built static hypergraphs from several data sets of time-resolved face-to-face human interactions, as in[30,32]. The data sets are provided by the SocioPatterns collaboration[46–48] and by the Contacts among Utah's School-age Population (CUSP) project[49] and describe interactions between individuals in several contexts: a hospital (LH10[85]), a workplace (InVS15[47,86]), a conference (SFHH[47]), a high-school (Thiers13[87]), two primary-schools (LyonSchool[88], Elem1[49]) and a middle-school (Mid1[49]). For these data sets we carried out an aggregation procedure to obtain static hypergraphs: (i) we aggregate the data over time windows of 15 minutes; (ii) we identify the cliques in each time window, i.e. groups of nodes forming a fully connected cluster, (iii) we identify in each temporal window the maximum cliques, i.e. cliques not completely contained in a larger clique, and promote them to a hyperedge status.

Finally, we consider hypergraphs built from ecological data sets provided by the Web of life: ecological networks database[59]. The data are in the form of bipartite graphs, where the nodes represent insect species or plants and the links connecting them represent a pollination relationship. Starting from these bipartite graphs we built two types of projected hypergraphs, obtained respectively by considering insect species as nodes and hyperedges connecting species that pollinate the same plant, or by considering plants as nodes and hyperedges connecting plants that are pollinated by the same insect species. Here we use two bipartite networks: M_PL_015[59,61] and M_PL_062[59,60], yielding the hypergraphs M_PL_015_ins and M_PL_062_ins with insects as nodes, and the hypergraphs M_PL_015_pl and M_PL_062_pl with plants as nodes.

Overall, the data sets considered describe interactions in several different environments, mediated by different mechanisms. They correspond to a wide variety of statistical properties (e.g. data set size, hyperedges size distributions), as shown in the SI where these statistical properties of the data sets are reported in detail.

## Hypergraph randomization procedure

Given a hypergraph $\mathcal{H}$, we generate a randomized realization $\mathcal{H}'$ with the same number of nodes $N$, the same number of hyperedges of each size $m$, $\forall m \in [2, M]$, and that also preserves the degree vector $\boldsymbol{d}(i)$ of each node $i$. Each realization is obtained through a hypergraph shuffling procedure analogous to those used in Refs. 12,89, which works as follows. At the beginning of the shuffling procedure $\mathcal{H}' = \mathcal{H}$; then we randomly select two hyperedges of the same size $m$, $e = \{i_1, i_2, \ldots, i, \ldots, i_m\}$ and $f = \{j_1, j_2, \ldots, j, \ldots, j_m\}$. We then randomly draw a node from each of

the two hyperedges, let us say respectively $i$ and $j$, and replace $e \rightarrow e' = \{i_1, i_2, \ldots, j, \ldots, i_m\}$ and $f \rightarrow f' = \{j_1, j_2, \ldots, i, \ldots, j_m\}$. The hyperedge swap is accepted if neither $e'$ nor $f'$ already existed in $\mathcal{H}'$. Note that the other hyperedges to which $i$ and $j$ belongs are not changed. The procedure is repeated $\forall\, m \in [2, M]$ until $10^5$ hyperedge swaps are performed for each $m$ (if there are at least 4 hyperedges of size $m$, otherwise the shuffling procedure is not applied for that $m$). The results presented in the manuscript following this procedure correspond to $10^3$ independent realizations of the shuffled hypergraphs.

## Models and stochastic simulations

**Higher-order nonlinear contagion.** We performed stochastic numerical simulations of the higher-order nonlinear contagion model on each empirical static hypergraph. The simulations are performed with discrete time-steps. The $S \rightarrow I$ infection mechanism is the same for the SIR and the SIS models: for each time-step $\Delta t$, given a hyperedge of size $m$ containing $i$ infected nodes, each of the susceptible nodes in it can be infected with probability $(1 - e^{-\lambda i^\nu})$. Therefore, the probability that a node $j$ is infected in a time-step $\Delta t$ is:

$$p_j = 1 - \prod_{e \in \mathcal{E}(j)} e^{-\lambda i_e^\nu}, \qquad (3)$$

where $\mathcal{E}(j)$ denotes the set of hyperedges in which the node $j$ is involved and $i_e$ is the number of infected nodes in the hyperedge $e$. Each infected node heals (returning susceptible in SIS or gaining immunity in SIR) with probability $\mu$ in each time-step.

In the SIS process, the population is initialized with a single infectious seed randomly selected in the population and the process is iterated until the system reaches a steady state with a fluctuating number of infectious. An observation time window $T$ is then considered and the time $\tau$ spent in the infectious state is estimated for all nodes over that time-window. The results are averaged over $10^3$ simulations.

In the SIR process the population is initialized with a single infectious seed $j$ and the dynamic process is iterated until no more infectious nodes are present: the final epidemic size $R_\infty(j)$ obtained by seeding the infection in $j$ is defined as the final number of nodes in the $R$ state. The results are averaged over 300 simulations for each infection seed $j$.

**Higher-order NG process.** We also performed numerical simulations of the higher-order NG process on the empirical hypergraphs. The system with $N$ nodes is initialized by fixing $N_p$ nodes as belonging to the committed minority (equivalently, with a fraction $p = N_p/N$ of committed nodes), with only the name $A$ in their dictionary, and setting the dictionaries of all the other nodes of the majority with only the name $B$. The committed nodes are selected following one of the three seeding strategies, i.e. randomly from the whole population or as the $N_p$ nodes with highest $s$-coreness or hypercoreness. If several nodes have the same coreness value, the committed nodes are randomly selected within the coreness class.

The simulations are performed in discrete time-steps: at each time-step a hyperedge is randomly selected (activation of the group) and within it a node is randomly chosen as the speaker, while the other nodes behave as listeners. The speaker randomly selects a name in their dictionary and all nodes in the group update their dictionary according to the chosen agreement rule (except for the committed nodes). The process is iterated until the system reaches the absorbing state where all nodes have only the name $A$ in their dictionary, i.e. $n_A(t) = n_A^* = 1$, or until the system has evolved for $t_{max}$ time-steps: in this last case the stationary fraction of nodes with the name $A$ in their dictionary $n_A^*$ is obtained by averaging $n_A(t)$ over 100 values sampled in the last $T = 50,000$ time-steps. The results refer to the median values obtained over 200 simulations.

## Reporting summary

Further information on research design is available in the Nature Portfolio Reporting Summary linked to this article.

## Data availability

The data that support the findings of this study are publicly available. The SocioPatterns data sets at http://www.sociopatterns.org/; the Contacts among Utah's School-age Population data sets at https://royalsocietypublishing.org/doi/suppl/10.1098/rsif.2015.0279; the online and political interactions data sets at https://www.cs.cornell.edu/~arb/data/; the Web of life ecological data sets at https://www.web-of-life.es.

## Code availability

The code is available at https://github.com/marco-mancastroppa/hypercore-decomposition/ and on Zenodo[90] at https://doi.org/10.5281/zenodo.8345106. The code uses the CompleX Group Interactions, XGI, Python library[91].

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

## Acknowledgements
M.M. and A.B. acknowledge support from the Agence Nationale de la Recherche (ANR) project DATAREDUX (ANR-19-CE46-0008). I.I. acknowledges partial support from the James S. McDonnell Foundation 21st Century Science Initiative Understanding Dynamic and Multi-scale Systems - Postdoctoral Fellowship Award.

## Author contributions
M.M., I.I., G.P., A.B. designed the study; M.M. performed the numerical simulations; M.M., I.I., G.P., A.B. analyzed the results; M.M. and A.B. wrote the first draft. M.M., I.I., G.P., A.B. contributed to the current draft.

## Competing interests
The authors declare no competing interests.
