## [Peer Review File · Nature Communications]

REVIEWER COMMENTS

Reviewer #1 (Remarks to the Author):

The authors propose a measure for k -cores in hypergraphs. The main feature, compared to the standard definition, is that they take into account also the size of the hyperedges that make the core, with an additional index m . This also implies that one has more than one number per node and a given k , as there is one for each m . Hence they consider a weighted average to get a scalar quantity as a centrality for each node (although they always consider weight equal to 1 in the experiments, i.e. a standard unweighted average).

They then show how the number of elements in the core varies in different datasets and how seeding a dynamical process (e.g. SIS, SIR) by this coreness centrality impacts the spreading. The main contribution of this work is proposing a metric that accounts for hyperedges size, and then exploring how this behaves in some real datasets. I am not convinced that this is enough for the broader scope of Nature Communications. Instead, this work looks more suited to a specialized journal. In particular, there are several works proposing similar ideas for k -coreness in hypergraphs as done in this paper, see for instance a recent arxiv:2301.06712. The notion of (k,m) coreness is also not novel, see Ahmed et al. 2007, Cerinsek and Batagelj 2015, Liu et al. 2020. I also do not see a clear advantage of using this metric over others, mainly because the authors did not make any comparison with other possible choices (beyond a naive random seeding choice in the last plot), so it is hard to tell what is peculiar and beneficial of this metric and what is maybe already intrinsic in simpler hypergraph properties as node degree and hyperedge size. The results from experiments are not surprising, nodes more connected and in larger hyperedges are more effective for spreading an epidemics. Perhaps I could have seen this also if I used some other metric based on node degree and/or hyperedge size, but I do not have a convincing comparison to gain a better idea.

I also did not fully understand the advantage of using the index m , compared to the standard k -core definition, is it because often nodes in a k -core are connected via many pairwise edges and we want to better account for larger ones? This also something that should have been better clarified.

Similarly, they propose a weighted average for the centrality, but then they only show results for weight equal to one. How would otherwise $g(m)$ impact the results? What is the advantage of using that definition if you only use a simple average?

Finally, conceptually, the definition of a subgraph \mathcal{I} leads to the possibility of introducing hyperedges e which may not be part of the input hypergraphs (if I understand correctly). These are edges that have only a subset of nodes in A and whose cardinality is at least m . While this may not be a problem if we treat these non-existing edges as "auxiliary" edges needed to compute the coreness, I would have expected the authors to at least comment that this subgraph (not sure it is appropriate to call it this way) is potentially unrealistic, in that it contains edges that were not observed.

In summary, while I find this work an interesting addition to the current literature on hypergraphs, I am not convinced of the fundamental novelty and the overall significance of the results presented in the manuscript. Hence, I think this paper is best suited for publication in a specialized journal.

Reviewer #2 (Remarks to the Author):

The author presents the interesting new concept of (k,m) -hypercore decomposition of an hypergraph and study the role of such structure on some dynamical processes defined on hypergraph. The subject is interesting and timing to the raising interest of the scientific community on higher-order structures, the paper is well written. I therefore support its publication provided some minor modifications.

1)The proposed results are interesting and the claims insightful, however they are based on the analysis of only two models, the epidemic process and the naming game, moreover they rely on numerical simulations and no attempts to provide an analytical study has been performed. I thus invite the authors to rephrase their conclusions by taking into account such limitations of their study.

2)Authors didn't discussed the algorithmic complexity of the method proposed to determine the (k,m) -hypercore decomposition. They considered hypergraphs with number of nodes ranging from few hundreds to few thousand and similar ranges for hyperedges sizes, I was thus wondering is those are limitations of the algorithm or if larger cases could be considered as well. Overall a discussion of the bottleneck of the algorithm would be worth for the reader.

3)Figures 2 & 3 compare results for hypergraphs of different sizes (in terms of m and k), maybe it would help the reader if the results were presented by using some normalisation for m and k , eg, m_{max} and k_{max} .

4)Once presenting the naming game, authors introduced "The population includes a committed minority of N_p individuals who do not obey these rules whenever they are listeners, but instead stick to their norm, a single name A (their dictionary is never updated)". In my opinion those agents could be named zealots as done in other works.

5) Some of the used hypergraphs originate from a directional process (eg email-US), did authors take into account such structure or did they transforme such data into an undirected structure?

6) In the randomisation procedure, once two nodes are swapped from the hyperedges e and f, are they also swapped from all the other possible hyperedges they belong to or just from those two?

Hyper-cores promote localization and efficient seeding in higher-order processes

(NCOMMS-23-15706-T)

Reply to the Reviewers' reports

M. Mancastropa¹, Iacopo Iacopini^{2,3}, Giovanni Petri^{2,4}, Alain Barrat¹

¹ Aix Marseille Univ, Université de Toulon, CNRS, CPT, Turing Center for Living Systems, Marseille, France

² Network Science Institute, Northeastern University London, London, E1W 1LP, United Kingdom

³ Department of Network and Data Science, Central European University, 1100 Vienna, Austria

⁴ CENTAI, Corso Inghilterra 3, 10138 Turin, Italy

July 4, 2023

Detailed reply to the report of Reviewer #1

COMMENT 1.1

The authors propose a measure for k -cores in hypergraphs. The main feature, compared to the standard definition, is that they take into account also the size of the hyperedges that make the core, with an additional index m . This also implies that one has more than one number per node and a given k , as there is one for each m . Hence they consider a weighted average to get a scalar quantity as a centrality for each node (although they always consider weight equal to 1 in the experiments, i.e. a standard unweighted average). They then show how the number of elements in the core varies in different datasets and how seeding a dynamical process (e.g. SIS, SIR) by this coreness centrality impacts the spreading. The main contribution of this work is proposing a metric that accounts for hyperedges size, and then exploring how this behaves in some real datasets. I am not convinced that this is enough for the broader scope of Nature Communications. Instead, this works looks more suited to a specialized journal.

Reply 1.1.—We thank the Reviewer for this summary of our work, however we respectfully disagree with the statement that our main contribution is to propose a metric and explore it in some datasets. Indeed, our work presents novel contributions along three lines: (i) we propose the (k, m) -core decomposition as a characterization tool, and show how it can uncover patterns in hypergraphs of very different nature and interest in different fields of research, in particular by comparing empirical decomposition of data sets with those of null models; (ii) we propose, as noted by the Reviewer, a new metric (in fact, a family of metrics), which takes into account the group structure and not only the structure of the projected network; (iii) we explore the interplay between the (k, m) decomposition and the new metric with several dynamical processes on hypergraphs, of different nature (spreading processes and dynamics of social conventions), showing that central cores play an important role, which depends on both parameters k and m ; we provide a comparison of the performance of our metric with centrality/coreness measures that do not take into account the hypergraph structure, but only the projection on a network.

We firmly believe that the combination of these points –expanded with respect to the previous version of our manuscript– makes our work of interest for the interdisciplinary audience of Nature Communications.

Action taken 1.1.—Prompted by the comments of the Reviewer, we have modified our manuscript as described below, to make our contribution clearer but mostly to add new investigations that reinforce our results. We hope that these modifications can convince the Reviewer of the relevance of our results for Nature Communications.

COMMENT 1.2

In particular, there are several works proposing similar ideas for k -coreness in hypergraphs as done in this paper, see for instance a recent arxiv:2301.06712. The notion of (k, m) coreness is also not novel, see Ahmed et al. 2007, Cerinsek and Batagelj 2015, Liu et al. 2020.

Reply 1.2.—We would first like to mention that the recent arxiv mentioned by the Reviewer was in fact published on arXiv several days after our own work, as the arXiv number of our work (2301.04235) shows. The Reviewer is right that the notion of (k, m) -core was introduced previously, in the context however of bipartite networks. We have added citations to these papers and described the difference with our own work: these earlier works do not present any interpretation in terms of hypergraphs. In addition, none of the cited works (even arxiv:2301.06712, which is in the context of hypergraphs) study the empirical result of the decomposition in hypercores as we do (introducing in particular the comparison with null models), nor propose a new centrality metric, nor consider the interplay between hypercores and dynamical processes.

Action taken 1.2.—We have added references and discussed the novelty of our work with respect to them.

COMMENT 1.3

I also do not see a clear advantage of using this metric over others, mainly because the authors did not make any comparison with other possible choices (beyond a naive random seeding choice in the last plot), so it is hard to tell what is peculiar and beneficial of this metric and what is maybe already intrinsic in simpler hypergraph properties as node degree and hyperedge size. The results from experiments are not surprising, nodes more connected and in larger hyperedges are more effective for spreading an epidemics. Perhaps I could have seen this also if I used some other metric based on node degree and/or hyperedge size, but I do not have a convincing comparison to gain a better idea. I also did not fully understand the advantage of using the index m , compared to the standard k -core definition, is it because often nodes in a k -core are connected via many pairwise edges and we want to better account for larger ones? This also something that should have been better clarified.

Reply 1.3.—In previous works such as St-Onge et al., Comm. Phys. 2022 and Iacopini et al., Comm. Phys. 2022, it has been shown that large groups are important for several dynamical processes on hypergraphs. Moreover, it can indeed be expected from works such as Kitsak et al, Nat. Phys. 2010, that connectivity and coreness are also important. However, it remains unclear how to uncover the most important nodes within large groups, or how the interplay between connectivity and group size can be exploited. This is a crucial point, as traditional network approaches would suggest targeting hubs (highest-degree nodes), but without considering the fact that they might participate into group interactions of different sizes –creating in this way a potential trade-off between targeting highly connected nodes vs nodes that are part of large groups. Here, we tackle this challenge by exploring how both indices k and m influence the spreading power jointly: for instance, in Figure 3 we see that the spreading power increases with m at fixed k , and increases with k at fixed m ; we show in the insets of the new Figure 3, for representative example values of these parameters, how the impact on spreading processes depends in non-trivial ways on both k and m . We propose a family of centrality measures that combine the relative centrality of nodes in the decomposition at all sizes m to uncover the most important nodes for dynamical processes. We compare the performance of these centralities with two centralities based on coreness that do not include information on group size, being computed on the network projection of the data sets.

Action taken 1.3.—We have modified Figure 3 to show the dependency on both indices k and m , to show how the behaviour vs k depends on m and vice-versa. We have added a new figure (Figure 4) to show the comparison with two coreness centralities computed without taking into account group sizes, for the case of the spreading processes. We have expanded the results also on the naming game to add comparisons between centrality measures that either do or do not take into account group sizes.

COMMENT 1.4

Similarly, they propose a weighted average for the centrality, but then they only show results for weight equal to one. How would otherwise $g(m)$ impact the results? What is the advantage of using that definition if you only use a simple average?

Reply 1.4.—We would like to thank the Reviewer for pointing this out. In the previous version of our work, we had indeed not taken advantage of the flexibility of the hypercoreness definition. Prompted by their suggestion, we now make it clearer that we can define a family of centrality measures, discuss how it can be used in specific cases, and consider two different cases. As previously, we consider on the one hand the *size-independent* hypercoreness: in this simplest version, all sizes are weighted equally, which can be a natural definition that does not depend on the data set nor on the dynamical process considered. We also consider the *frequency-based* hypercoreness, which is the most natural way to insert a data-driven element into this centrality: indeed, each size is here weighted by its relative abundance in the data. We show how both these centralities outperform the network-based coreness ones in finding the most relevant nodes for spreading processes, and provide powerful seeding strategies for committed individuals in the emergence of social conventions.

Action taken 1.4.—We have updated the presentation of the hypercoreness family of centralities, and we present results obtained with two different versions of the hypercoreness centrality.

COMMENT 1.5

Finally, conceptually, the definition of a subgraph \mathcal{I} leads to the possibility of introducing hyperedges e which may not be part of the input hypergraphs (if I understand correctly). These are edges that have only a subset of nodes in A and whose cardinality is at least m . While this may not be a problem if we treat these non-existing edges as “auxiliary” edges needed to compute the coreness, I would have expected the authors to at least comment that this subgraph (not sure it is appropriate to call it this way) is potentially unrealistic, in that it contains edges that were not observed.

Reply 1.5.—The Reviewer is right. We have added a comment on this point. Hyperedges in \mathcal{I} might not be in \mathcal{H} , but they can still be interpreted as existing interactions if one assumes that subsets of a set of interacting nodes are indeed interacting (Note that this would be automatically the case if we consider simplicial complexes instead of the more general hypergraphs). In any case, our focus is on hypercores as sets of nodes rather than on the remaining hyperedges that connect these nodes, therefore this point does not impact our results.

Action taken 1.5.—We have added a comment on this point.

Detailed reply to the report of Reviewer #2

COMMENT 2.1

The author presents the interesting new concept of (k,m) -hypercore decomposition of an hypergraph and study the role of such structure on some dynamical processes defined on hypergraph. The subject is interesting and timing to the raising interest of the scientific community on higher-order structures, the paper is well written. I therefore support its publication provided some minor modifications.

Reply 2.1.—We are grateful for these very positive comments and the suggested modifications, which have helped us improve the manuscript.

Action taken 2.1.—We have taken all comments into account and modified our manuscript as detailed below.

COMMENT 2.2

The proposed results are interesting and the claims insightful, however they are based on the analysis of only two models, the epidemic process and the naming game, moreover they rely on numerical simulations and no attempts to provide an analytical study has been performed. I thus invite the authors to rephrase their conclusions by taking into account such limitations of their study.

Reply 2.2.—We agree with the Reviewer that there exist many other interesting dynamical processes on hypergraphs, and that our study has been limited to two types of processes.

Action taken 2.2.—We have added a comment in the discussion section to mention these limitations.

COMMENT 2.3

Authors didn't discuss the algorithmic complexity of the method proposed to determine the (k,m) -hypercore decomposition. They considered hypergraphs with number of nodes ranging from few hundreds to few thousand and similar ranges for hyperedges sizes, I was thus wondering if those are limitations of the algorithm or if larger cases could be considered as well. Overall a discussion of the bottleneck of the algorithm would be worth for the reader.

Reply 2.3.—We have considered publicly available data sets corresponding to interactions in very different contexts, and did not limit ourselves particularly in terms of sizes. The Reviewer is in any case right that the scalability of a method is always an important point, and we had not touched this point in our manuscript.

Action taken 2.3.—We have added a comment about how the decomposition scales with respect to the sizes of the data sets, and a corresponding section in the SI.

COMMENT 2.4

Figures 2 & 3 compare results for hypergraphs of different sizes (in terms of m and k), maybe it would help the reader if the results were presented by using some normalisation for m and k , eg, m_{max} and k_{max} .

Reply 2.4.—As all figure panels have the same sizes, the figures would in fact not be modified. If we instead normalize by the same value for all data sets, some figures would become unreadable, as the various data sets correspond to very different maximal values of k and m .

Action taken 2.4.—We prefer to keep the axes unmodified, as the reader can then see directly on the axis what are the

maximal values in each data set.

COMMENT 2.5

Once presenting the naming game, authors introduced “The population includes a committed minority of N_p individuals who do not obey these rules whenever they are listeners, but instead stick to their norm, a single name A (their dictionary is never updated)”. In my opinion those agents could be named zealots as done in other works.

Reply 2.5.—We agree with the Reviewer. The terms “committed” and “zealots” have been used by various authors and in various works to indicate the same concept.

Action taken 2.5.—We have added a comment to the fact that the committed individuals can also be called “zealots” and added references to articles in which this term is used.

COMMENT 2.6

Some of the used hypergraphs originate from a directional process (eg email-US), did authors take into account such structure or did they transform such data into an undirected structure?

Reply 2.6.—We have indeed transformed the data into undirected structures, when the original data was directed, which is here the case only for the two email data sets.

Action taken 2.6.—We have made this clear in the manuscript.

COMMENT 2.7

In the randomisation procedure, once two nodes are swapped from the hyperedges e and f , are they also swapped from all the other possible hyperedges they belong to or just from those two?

Reply 2.7.—The nodes are swapped only from the two selected hyperedges.

Action taken 2.7.—We have made this point clear.

REVIEWER COMMENTS

Reviewer #1 (Remarks to the Author):

The authors have responded to most of my concerns, the paper is now greatly improved.

While this is a solid paper, my main concern is still valid, as I am not fully convinced about the novelty of the measure and thus the contribution seems more incremental. It is true that the metric was originally proposed in the context of bi-partite graphs, but this can also be seen as one of the possible representation of hypergraphs. In fact, a recent arxiv 2301.08440, shows that this bi-partite k-core metric is equivalent to that of hypergraphs and also mentions that this definition was indeed introduced more explicitly in the context of hypergraphs in Limnios 2021. Hence I am not fully convinced that the metric that the authors propose here is new, as they claim.

In general I find this a solid work and relevant in showing how a k-core metric works in different datasets. My impression is

that, given the proposed metric was presented as a main innovation of this work by the author, and I am not sure of its novelty, I feel that the authors made an incremental step along previous works about k-core in hypergraphs. The

work should thus be published in some journal, but I am not sure this should be Nature Communications; for this, the paper should have a fundamental

contribution. However, I am not justified by the authors' revision that

their work meets this criterion.

Reviewer #2 (Remarks to the Author):

Authors replied to all my questions. The only still remaining point is the discrepancy between the generality of the main claim and the use of only two models to support it without providing any analytical study nor a motivation of the chosen models. Maybe a simple linear model, such as the one in [28] could have been used to understand the role of the new metric. I thus invite the authors to rephrase their conclusions by taking into account the limitations of their study and try to get some analytical results, even if on a linear model.

Detailed reply to the report of Reviewer #1

COMMENT 1.1

The authors have responded to most of my concerns, the paper is now greatly improved. While this is a solid paper, my main concern is still valid, as I am not fully convinced about the novelty of the measure and thus the contribution seems more incremental. It is true that the metric was originally proposed in the context of bi-partite graphs, but this can also be seen as one of the possible representation of hypergraphs. In fact, a recent arxiv 2301.08440, shows that this bi-partite k-core metric is equivalent to that of hypergraphs and also mentions that this definition was indeed introduced more explicitly in the context of hypergraphs in Limnios 2021. Hence I am not fully convinced that the metric that the authors propose here is new, as they claim.

Reply 1.1.—We thank the Reviewer for recognizing that our manuscript has improved. We respectfully disagree however with the statement that our work is incremental, and think that it comes from a misunderstanding of what is the novelty of our work.

First, we agree with the Reviewer about the equivalence of the decomposition with the case of bipartite graphs, and indeed we had already stated this explicitly.

However, we believe there is a confusion about the term “metric”: the metric we introduce as hypercoreness is indeed new. In the arXiv mentioned by the Reviewer, which came out *after* our own work, the authors mention a t-hypercoreness, which is however a function and thus not a metric, and corresponds to what we call shell index. As we explain in the manuscript, it is impossible to rank nodes by comparing these functions, so that we adopt another strategy by transforming these functions into scalars, the g-hypercoreness R_g . This allows then to rank nodes.

Finally, we would like to emphasize that the novelty of our work is threefold: (i) we apply the decomposition to a variety of empirical hypergraphs, suggest to compare it with a suitable null model and show that it provides a fingerprint of the data; (ii) we provide a new family of centrality measures, the hypercoreness; (iii) we investigate the role of central hypercores and of nodes with high hypercoreness in several dynamical processes on hypergraphs.

Action taken 1.1.—We have modified the abstract and introduction to make clearer the link with the bipartite case and to better highlight what constitutes the novelty of our work.

COMMENT 1.2

In general I find this a solid work and relevant in showing how a k-core metric works in different datasets. My impression is that, given the proposed metric was presented as a main innovation of this work by the author, and I am not sure of its novelty, I feel that the authors made an incremental step along previous works about k-core in hypergraphs. The work should thus be published in some journal, but I am not sure this should be Nature Communications; for this, the paper should have a fundamental contribution. However, I am not justified by the authors' revision that their work meets this criterion.

Reply 1.2.—We once again emphasize that the decomposition in itself is not what constitutes novelty in our work. The “metric” hypercoreness however is novel (while the Reviewer seems to suggest as metric the decomposition itself, which is not our claim) and is one element of novelty, but, most importantly and as highlighted above, the novelty of our work is actually threefold and not limited to a novel definition. Therefore, and given the three elements of novelty described above, we strongly believe that our work is not incremental, but it brings instead crucial new elements for the study of hypergraphs.

Action taken 1.2.—We have modified the abstract and introduction to better highlight what constitutes the novelty of our work.

Detailed reply to the report of Reviewer #2

COMMENT 2.1

Authors replied to all my questions. The only still remaining point is the discrepancy between the generality of the main claim and the use of only two models to support it without providing any analytical study nor a motivation of the chosen models. Maybe a simple linear model, such as the one in [28] could have been used to understand the role of the new metric. I thus invite the authors to rephrase their conclusions by taking into account the limitations of their study and try to get some analytical results, even if on a linear model.

Reply 2.1.—We thank the Reviewer for recognizing we replied to their questions. We would like to note that our work considers two types of dynamical processes but overall 6 models and not only 2: indeed, we consider a consensus formation process and two models of spreading with different ways of taking into account higher-order effects, and both SIS and SIR models in each case.

We agree with the Reviewer that our work has limitations, and we have made them clearer and discussed more in the discussion section. We have also added motivations for the chosen models. These indeed have been well studied recently, and have shown their relevance in terms of new phenomenology driven by higher order effects. Moreover, the Naming Game has been shown to reproduce well some controlled experiments on consensus formation. We have thus chosen these models as paradigmatic and of interest when studying higher-order dynamics on hypergraphs.

We of course also agree that it would be great to have analytical results. In general, the linearization suggested by the Reviewer is a good approach to study the stability of dynamical processes on hypergraphs [28] (now [31]). However, it has been recently shown that non-linearities play a crucial role in higher-order dynamical processes on hypergraphs, suggesting that a linear model may not contain enough structure to accurately approximate the full dynamics (e.g., Neuhäuser et al., 2022, and arxiv 2306.01813, 2023). We also emphasize that, even in the case of usual networks, analytical results on the spreading power of nodes and on finding metrics to determine the most influential nodes are very scarce. For instance, in the review paper Pastor-Satorras et al, 2015, it is written "*a clear picture that uniquely determines the best centrality measure that identifies superspreaders for different epidemic models and different networks has yet to emerge*". Some works use mean-field approaches, focusing mostly just on the degree centrality, and results are usually obtained from heuristic and numerical approaches (e.g., Erkol et al, 2019, compare 16 heuristic approaches on a corpus on empirical networks). We are currently not aware of any existing analytical approach that could deal with concepts of coreness and highlight its role. With these premises, devising an analytical treatment for the current study remains out of reach.

Action taken 2.1.—We have added several comments to motivate the models considered, and also in the discussion section to better state the limitations of our work.

REVIEWERS' COMMENTS

Reviewer #1 (Remarks to the Author):

The authors have better clarified what they were proposing and what was there in the literature. I think that a reference to Limnios 2021 is still missing (it is correct that the arxiv 2301.08440 came after the authors' work, but the concept of hypercoreness was already mentioned in Limnios in 2021; the arxiv I mentioned is for the connection with bi-partite networks).

The paper is scientifically sound and clear to read and now it is clearer what it is that they are proposing, the paper has improved from the previous draft.

While we may still have disagreement in the fact that what they define as threefold novelty may or may not be enough for Nature Communications, I prefer to leave this as a matter for the editor to decide.

Reviewer #2 (Remarks to the Author):

I'm ok with the last version of the manuscript. The changes made answered to the points I raised.

Detailed reply to the report of Reviewer #1

Comment 1.1

The authors have better clarified what they were proposing and what was there in the literature. I think that a reference to Limnios 2021 is still missing (it is correct that the arxiv 2301.08440 came after the authors' work, but the concept of hypercoreness was already mentioned in Limnios in 2021; the arxiv I mentioned is for the connection with bi-partite networks). The paper is scientifically sound and clear to read and now it is clearer what it is that they are proposing, the paper has improved from the previous draft. While we may still have disagreement in the fact that what they define as threefold novelty may or may not be enough for Nature Communications, I prefer to leave this as a matter for the editor to decide.

Reply 1.1.—We are grateful to the reviewer for these comments. We have added the reference to Limnios 2021 to correctly acknowledge that this concept was already mentioned there.

Action taken 1.1.—We have added the reference (reference [40]) in several points of the manuscript.